# GROUP EQUIVARIANT CONVOLUTIONAL NETWORKS

## ABSTRACT

Convolutional Neural Networks (CNN) are using symmetry priors to make the best out of the properties of the data, in particular translation invariance in images. Group Equivariant CNN (Cohen & Welling, 2016) extend CNN by using invariances from other groups of symmetry. After exploring their mathematical properties, we confirm their performances on Rotated MNIST and CIFAR10, introduce some extensions with GAN and propose applications in High Energy Physics.

## 1 INTRODUCTION

Bronstein et al. (2021) aim to unify deep learning strategies under the *Geometric Deep Learning* denomination, by approaching geometry as the study of invariants, i.e. properties unchanged under some group of transformations, called the symmetries of the geometry. CNN classicaly make use of translation invariance and self-similarity across a classic image domain. Some domains have other priors we would like to make use of, like rotation invariance in molecules. From this observation, Cohen & Welling (2016) introduce Group Equivariant CNN, using the $p4$ and $p4m$ group transformations.

In this paper, we first show their equivariance. We then present our performances on our implemented and optimized models from Cohen & Welling (2016), on Rotated MNIST. We also propose a new model, P4MCNN, which is P4CNN with $p4m$ convolutional layers. We took the same approach on the All-CNN-C structure (Springenberg et al., 2014), computing its $p4$ and $p4m$ variants on CIFAR 10. In section 3.3, we experiment Group Equivariant GAN (Dey et al., 2020) using $p4$ convolutions in the discriminator. Finally, in section 4, we propose applications for Group Equivariant CNN and GAN to the Higgs Boson ML challenge.

## 2 GROUP EQUIVARIANT NETWORKS

Group Equivariant Networks are made of 3 types of layers: G-correlation, G-pooling, nonlinearity. We need to prove that each operation is equivariant i.e commutes with G-transformations of the domain of the image, for the G-CNN to enforce group equivariance.

A **G-equivariant correlation** ($\star$) is a variant of convolution, in which we replace the shift by a more general transformation from some group G. See appendix A for details on (1).

$$[f \star \psi](g) = \sum_{y \in \mathbb{Z}^2} \sum_{k=1}^{K^l} f_k(y)\psi_k(g^{-1}y) \tag{1}$$

G-correlation is equivariant i.e. $[L_u f] \star \psi = L_u[f \star \psi]$ where $L_u$ is the operator for a transformation $u$. G-correlation preserves the transformation of the previous layer but if G is not commutative, neither the G-correlation. The set of G-equivariant feature maps is closed for addition. However, we can add only one bias term per G-feature map in order to preserve equivariance.

**Pooling** commutes with $L_u$. For more explanations on subgroup and coset poolings, see**B**. **Applying a nonlinearity** $\nu : \mathbb{R} \to \mathbb{R}$ to the feature map is equivalent to apply a composition operator $C_\nu$, that acts on functions by post-composing them with $\nu$. As the left operator $L$ acts by pre-composition, $C$ and $L$ commute: the transformation properties of the previous layer are transfered to the feature map.

# 3 EXPERIMENTS

## 3.1 ROTATED MNIST

The models were trained on Rotated MNIST using Pytorch, and tested on both MNIST and Rotated MNIST (see 3.3). P4CNN is indeed the best structure from the original paper, it performs better without dropout and without the last max pooling layer (P4CNN drop and no drop) than with it (P4CNN max). We also found that SGD (lr=0.01, momentum=0.5) was a better optimizer than Adam. The new P4MCNN is the best model, reaching $96\%$ of accuracy on both datasets. All in all, G-CNN are proven useful even on datasets with no $p4$ symmetries, as they perform well on MNIST.

## 3.2 CIFAR10

We worked on All-CNN-C (Springenberg et al., 2014). With its 9 convolutional layers (see **C**), even a CUDA GPU from Google Colab could not handle the training on our computer. We still implemented it and its $p4$ and $p4m$ variants (Cohen & Welling, 2016).

## 3.3 GROUP EQUIVARIANT GAN

We trained the Vanilla GAN (Rath, 2020) on Rotated MNIST on a GPU from Google Colab (see3.3), however we could not do the same with P4CNN as the discriminator, due to an incompatibility of the chainer module with the CUDA version. Nevertheless, this idea is promising as it was shown that using p4-convolutions improves both the mean and minimum Fréchet distance, that captures sample fidelity and diversity (Dey et al., 2020).

| Model | Rotated MNIST | MNIST |
|---|---|---|
| Z2CNN | 87% | 90% |
| P4CNN RP | 92% | 93% |
| P4CNN (max) | 94% | 95% |
| P4CNN (drop) | 94% | 95% |
| P4CNN (no drop) | 95% | 96% |
| P4MCNN | 96% | 96% |

Table 1: Accuracy performances on two test sets when trained on Rotated MNIST

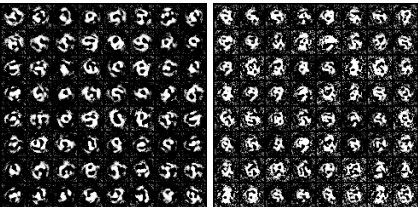

Figure 1: Epochs 88 and 197 of the Vanilla GAN

# 4 GENERALIZATIONS AND APPLICATIONS

Group Equivariant CNN and GAN are particularly interesting for problems with data symmetries, for example the famous Higgs Boson ML challenge (url). The data consists of simulated particle collisions, generated to mimic those expected at the LHC. Due to symmetries of the detectors, we have class-invariant symmetries in the data: to flips in the transverse or longitudinal planes, and to rotations in the azimuthal angle (Strong, 2020).

Strong (2020) used an ensemble of simple NN , so we could use Group Equivariant layers adapted to the problem to upgrade them. Moreover, as the data used is already generated, we could generate more samples using Group Equivariant GAN, to help the model generalize. This could be a new way to tackle the complexity of this data, as an alternative to *data-fixing* (aligning the events, section 4.5.2) or an extension to simple *data augmentation* (section 4.5.3) (Strong, 2020).

# 5 CONCLUSION

Group Equivariant CNN perform better than classic CNN on simple datasets, and pave the way for more group based neural networks that could improve our methods in High Energy Physics and in problems with data symmetries in general (3D molecules, etc).

ACKNOWLEDGEMENTS

I would like to thank Emanuele Rodolà, professor of Computer Science at Sapienza University of Rome, for his *Deep Learning and Applied AI* 2022 class I followed, and for pushing his students to participate to Tiny Papers.

URM STATEMENT

The author acknowledges that they meet the age and gender URM criteria of ICLR 2023 Tiny Papers Track.

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

## APPENDIX A  G-CORRELATION

For the first layer of the network, both $f$ and $\psi$ are functions of $\mathbb{Z}^2$, but $f \star \psi$ is a function on the group G. Then, for the next layers, the filters $\psi$ must be functions on G, so the first sum is now over G :

$$[f \star \psi](g) = \sum_{y \in \mathbb{Z}^2} \sum_{k=1}^{K^l} f_k(y)\psi_k(g^{-1}y) \tag{2}$$

## APPENDIX B  SUBGROUP POOLING AND COSET POOLING

Pooling can be divided into two steps, the pooling without stride, and the subsampling.

Doing a pooling with stride is equivalent to sub-sample the pooled feature map on a subgroup $H \subset G$:this subsampled feature map is equivariant to H but not to G.

We can obtain full G-equivariance by choosing our pooling region U to be a subgroup $H \subset G$. The resulting pooling domains $gH$ are called (right-hand) *cosets* in group theory.

Cosets are similarly invariant ($ghH = gH$), so we can arbitrarily choose one coset representative per coset to sample on, and cosets issued from the same group $H$ have no elements in common. It means the cosets partition the group into non-overlapping regions.

This is the notion of equivalence classes: the feature map is a function on the quotient space $G/H$, and two transformations are considered equivalent if they belong to the same coset.

## APPENDIX C    ALL-CNN-C

To be trained, the structure of All-CNN-C requires GPU ressources we did not have: on CIFAR 10, it would have required approximately 10 hours on a GPU according to Springenberg et al. (2014).

| All-CNN-C |
| --- |
| $3 \times 3$ conv. 96 ReLU |
| $3 \times 3$ conv. 96 ReLU |
| $3 \times 3$ conv. 96 ReLU with stride $r = 2$ |
| $3 \times 3$ conv. 192 ReLU |
| $3 \times 3$ conv. 192 ReLU |
| $3 \times 3$ conv. 192 ReLU with stride $r = 2$ |
| $3 \times 3$ conv. 192 ReLU |
| $1 \times 1$ conv. 192 ReLU |
| $1 \times 1$ conv. 10 ReLU |
| global averaging over $6 \times 6$ spatial dimensions |
| 10 or 100-way softmax |

Figure 2: All-CNN-C architecture

