# OpenReview forum: "Group Equivariant Convolutional Networks"
_ICLR.cc/2023/TinyPapers — Submitted to Tiny Papers @ ICLR 2023_

### Official Review · Reviewer_qEtd · 2023-03-19

**Confidence:** 4

**Summary Of Contributions:**

This paper explores some group equivariant CNN architectures, motivated by problems in HEP. All experiments are performed on the rotated-MNIST dataset.

**Rating:**

Needs Clarification (NC): a submission which does not meet the reviewing criteria and needs clarification for its described problem or solution

**Strengths And Weaknesses:**

This paper addresses an interesting topic, and the motivation for equivariance as a useful inductive bias for HEP data is clear. However, I think it requires clarification. I assess the submission with respect to each of the reviewing criteria below; I then suggest specific changes to address each of these points in the "Suggested Changes" section.

- **Clarity**: I found it difficult to read the submitted manuscript. Many definitions are missing (e.g., what is the group $p4m$?), and it is sometimes hard to differentiate between which ideas are taken from prior work and which are the author's contributions.

- **Correctness**: The rotated-MNIST results seem sound, but I had difficulty judging the correctness of the results given the lack of experimental details (c.f. "Reproducibility").

- **Reproducibility**: The paper does not provide code, nor does it include adequate details regarding experimental setup. Please see specific suggestions below.

- **Follows basic requirements**: The paper fits within the 2-page limit, but contains a non-anonymous acknowledgement.

**Suggested Changes:**

1. The author should include an explicit statement of what the contribution of this paper is. From the submitted paper, it is not clear whether the rotated-MNIST results are known from previous works. Confirmatory results are valuable, but they should be framed as such. This would also make it easier to judge the correctness of the manuscript. In that vein, the potential application to HEP seems to be a review of previous work, rather than a new perspective.

2. The experiment section would be easier to read if the author provided a short summary of what they aim to show at the beginning of the section. As it stands, it is not clear how the different results fit together.

3. To enable reproducibility, it would be useful to either provide code or to include in the Appendix a detailed description of architectural and training details. As it stands, Section 3.1 mentions experiments with different optimizers and different parameters, but does not actually give their results. It is thus difficult to judge whether the claims made there are correct.

4. The goal of Section 2 should be stated more clearly. I was not able to discern precisely what is being claimed vis-a-vis previous work or the definition of a G-CNN, a confusion which largely stems from my overall lack of clarity regarding what the contribution of the manuscript is.

---

### Official Review · Reviewer_6MGY · 2023-03-22

**Confidence:** 4

**Summary Of Contributions:**

This paper presents initial work towards extending group equivariant CNNs towards GANs and physics applications. The initial experiments implement previously proposed classification architectures and compare results across test datasets and hyperparameter modifications. Further extensions are proposed, towards group equivariant GANs and their application for generating additional data of particle collisions.

**Rating:**

Great Start (GS): a submission which meets some of the reviewing criteria but has room for improvement

**Strengths And Weaknesses:**

Strengths:
* The proposed ideas are interesting.
* The claims made regarding working experiments are justified.
* The submission adheres to formatting requirements and mostly follows the ICLR code of conduct (see anonymity tip below).

Weaknesses:
* The results of the paper are not very clear. Not many claims are made nor are many results presented, and neither are discussed and analyzed. The included results overlap with Cohen & Welling (2016) (both the paper and Github repository).
* Code is not provided.

**Suggested Changes:**

The results in Sections 3.1 and 3.3 are a great start, but should be discussed much more thoroughly, as these are the key contributions. Why could *p4* and *p4m* equivariance be beneficial for both test datasets? What does the transfer from RotatedMNIST to MNIST tell us? What does Figure 1 tell us and why? How do the contributions in Section 31. differ from Cohen & Welling (2016) and the proposals in Section 3.3 differ from Dey et al. (2020)?

Rather than mention things that did not work implementationally in the main paper, focus on the experiments that did work. Sections 3.2, 3.3, and 4 could all be reduced by mentioning these ideas as "future work" within Section 4, yielding much more space for discussing the findings from Section 3.1.

Regarding All-CNN-C, note that training efficiency has improved significantly since 2014, so previously reported runtimes may no longer be relevant. For a more accessible experiment, the channel widths or overall depth could have been reduced, or a dataset that is more "natural" than MNIST yet still less expensive than CIFAR10, such as FashionMNIST, could have been used.

Refer to the specific table or figure rather than the section it is located in. Also, always name the object being referenced: i.e. "For more explanations on subgroup and coset poolings, see Appendix B" and " tested on both MNIST and Rotated MNIST: the results are shown in Table 1".

Make sure to explicitly define all of your models. This can be done by giving the base architecture for P4CNN in an appendix, and then describing how each variant differs. For understandability and reproducibility, a reader should be able to reimplement your architectures. These definitions are also crucial for ensuring the models are fairly compared: for example, between Z2CNN and P4CNN, did you equate number of independent channels, total number of parameters, or some other structural aspect?

Define all new notations, variables, and names: $K^l$, "RP", and "Z2CNN" seem to come from Cohen & Welling (2016) but this is not clear to the reader.

Appendix A is rather unclear: Equation 2 is identical to Equation 1, but the text introducing it implies that it should be different (particularly changing the first summation). Perhaps you meant to distinguish between lifting and group convolutional layers?

Acknowledgements should generally be left out of the double blind submission to avoid loss of anonymity.

---

### Meta-Review · Area_Chair_JMZJ · 2023-04-04

**Recommendation:** Invite to revise
**Confidence:** 3

**Metareview:**

This paper tackles an interesting problem. Changes, however, can be made to improve its clarity. Specifically, I'm not sure what the primary claim of the paper is.

**Summary:**

This paper surveys group equivariant convolutional networks.

**Reason For Not Giving A Higher Recommendation:**

Unclear main claim and contributions.

**Reason For Not Giving A Lower Recommendation:**

NA

---

### Decision · Program_Chairs · 2023-04-10

No revision received; not invited to archive